# Ormona® SI and Ormona® RC—New Nutraceuticals with Geranylgeraniol, Tocotrienols, Anthocyanins, and Isoflavones—Decrease High-Fat Diet-Induced Dyslipidemia in Wistar Rats

Ana Paula Santos Rodrigues [1,2], Raimundo da Silva Barbosa [1], Arlindo César Matias Pereira [3],
Mateus Alves Batista [4], Priscila Faimann Sales [1,2], Adriana Maciel Ferreira [1], Nayara Nilcia Dias Colares [1],
Heitor Ribeiro da Silva [1], Marcelo Oliveira dos Santos Soares [1], Lorane Izabel da Silva Hage-Melim [4]
and José Carlos Tavares Carvalho [1,2,5,*]

[1] Laboratory of Drugs Research, Biology and Healthy Sciences Department, Pharmacy Faculty, Federal University of Amapá, Macapá 68902-280, Brazil
[2] Programa de Pós-Graduação em Inovação Farmacêutica, Departamento de Ciências Biológicas e da Saúde, Colegiado de Farmácia, Universidade Federal do Amapá, Macapá 68902-280, Brazil
[3] School of Pharmaceutical Sciences of Ribeirão Preto, University of São Paulo (FCFRP-USP), São Paulo 14040-903, Brazil
[4] Laboratory of Pharmaceutical and Medicinal Chemistry (PharMedChem), Federal University of Amapá, Macapá 68902-280, Brazil
[5] University Hospital of the Federal University of Amapá, Macapá 68902-336, Brazil
[*] Correspondence: farmacos@unifap.br or jose.tavares.1@ebserh.gov.br

**Abstract:** Dyslipidemia is a metabolic condition linked to increased morbidity. This study aimed to evaluate the effect of two new nutraceuticals derived from annatto (*Bixa orellana*), açaí (*Euterpe oleracea*), and soybean germ or red clover isoflavones (*Trifolium pratense*) against high-fat-induced dyslipidemia in female Wistar rats. The formulations were assessed through GC-MS and HPLC-UV/Vis. Next, female Wistar rats received daily administrations of coconut oil and were treated with Ormona® SI (OSI), Ormona® RC (ORC), soybean germ (SG), simvastatin (simv.), simvastatin + Ormona® SI (simv. + OSI), or only distilled water (control). Biochemical tests were performed using the animals' blood, and their arteries were screened for atheroma formation. The results show abnormal TC, TG, LDL, and HDL levels caused by the high-fat diet, increased glucose levels, hepatotoxicity, nephrotoxicity, and atherogenesis. The nutraceuticals significantly ameliorated all conditions, with results that are similar to the group treated with simvastatin. Notably, the groups treated with either Ormona® or simvastatin + Ormona® had better liver protection than those treated only with SG or simvastatin; additionally, the nutraceuticals could prevent atherogenesis, unlike SG. The results indicate a high efficacy of the nutraceuticals in preventing dyslipidemia and its complications.

**Keywords:** dyslipidemia; anthocyanins; isoflavones; tocotrienols; geranylgeraniol

## 1. Introduction

Dyslipidemia is characterized by abnormal blood lipid levels, such as increased low-density lipoproteins (LDL), decreased high-density lipoproteins (HDL), increased total triglycerides (TG), and increased total cholesterol (TC). These abnormalities are linked to a higher risk of cardiovascular diseases. The current standard treatment for dyslipidemia is statins, which are known to reduce both cardiovascular events and mortality [1,2].

However, despite the unquestionable usefulness of statins, not all patients can take them due to intolerance and myotoxicity. Hence, other approaches are being developed to treat the condition, including lipid-lowering nutraceuticals. In this context, annatto (*Bixa orellana*) is a plant species with known hypocholesterolemic activity, which is due at least in part to the tocotrienols in its composition and geranylgeraniol [3–6].

Ormona® SI and RC are granules derived from Chronic®, a nutraceutical composed of granules from a standardized extract of *B. orellana*. Our group showed that the treatment with this product could remarkably improve the blood-lipid profile and decrease bone calcium loss in orchiectomy-induced testosterone deficiency in male rats, with overall better results than the oil, which was attributed to being due to pharmacokinetics difference [6].

In Ormona SI® (OSI) and Ormona RC® (ORC) there is an addition of standardized extracts of soybean germ (*Glycine max*) or *Trifolium pratense*, respectively – whose major markers are isoflavones. Isoflavones are known for their health benefits and lipid-lowering capacity [7], mainly for menopausal women, as demonstrated in controlled clinical studies [8]. Additionally, there is an addition of a standardized extract of açaí (*Euterpe oleracea*) in both formulations, which is a known source of anthocyanins, mainly cyanidin-3-glucoside [9].

Considering the information given, the objective of this study was to evaluate the effect of these three nutraceuticals—Chronic®, Ormona® SI, and Ormona® RC—against dyslipidemia and its complications induced by a high-fat diet in Wistar rats.

## 2. Materials and Methods

### 2.1. Animals and Ethical Aspects

This study used female Wistar rats (*Rattus novergicus* albinus) acquired from the Animal Investigation Multidisciplinary Center at the University of Campinas (UNICAMP). The animals were kept in ventilated cages with a controlled temperature ($25 \pm 2$ °C), light/dark cycle (12/12 h), and diet.

This study followed the recommendations of the Brazilian College of Animal Experimentation (COBEA) and is in line with the resolution from the Veterinary Medical Board and the laws of the country concerning animal experimentation. The study was submitted and approved by the Animals Use in Research Ethics Committee (CEUA) from the Federal University of Amapá under the number 003/2021.

### 2.2. Material Test

The nutraceuticals products—Ormona SI® and Ormona RC®—were kindly provided by the company Ages Bioactive Compounds (Lot. 001, 2021 and 003, 2021). They were assessed for their δ-tocotrienol and geranylgeraniol content using gas chromatography coupled to mass spectrometry (GC-MS) and isoflavones (HPLC). A detailed composition profile is available in the patent process BR 020210174935. Ten percent of the formulations were composed of *Euterpe oleracea* (açaí) extract, which were previously characterized by our group [9].

### 2.3. Chemical Analysis

2.3.1. Tocotrienol and Geranylgeraniol Content Assessment

The tocotrienol and geranylgeraniol content was evaluated in the annatto part of the formulation. GC-MS was performed using Shimadzu equipment (GC 2010) with an auto-injector (AOC 5000), mass detector (70 eV, MS2010 Plus), and a fused silica column DB-5MS 30 m × 0.25 mm with 0.25 μm particle size (Agilent Advanced J&W). The split ratio was 1:30; helium was used as the carrier gas (65 kPa), and the injection volume was 1 μL. The injector and detector temperatures were set at 250 °C. The initial temperature of the column was set at 60 °C for 1 min., then heating was increased at 3 °C per min. until 290 °C.

The compounds were identified through their retention index by interpolating the retention time of a mixture of aliphatic hydrocarbonates (C9–C30) analyzed in the same conditions. The fragmentation pattern was also compared with the equipment library of mass spectra (NIST 5.0).

2.3.2. Isoflavones Quantification

A qualitative and quantitative isoflavone content analysis was performed with the isoflavones fractions from OSI, ORC, and SG through HPLC-UV/Vis (Shimadzu) using

the methodology described by de Melo et al. [10], with minor adaptations. The isoflavones were extracted through acid hydrolysis, liberating the aglycones from the glycosylated isoflavones. One hundred mg of the product was mixed with 40 mL of a solution of bi-distilled water acidified with 1% acetic acid and HPLC grade acetonitrile (4:6).

This solution was agitated in a magnetic stirrer for 60 min in the dark, under ambient temperature. Next, it was filtered with filter paper and a PVDF membrane 0.45 μm (Millex HV Durapore®). All the analytical standards (daidzein, glycitein, genistein, genistin) had 98% purity (Sigma-Aldrich®, São Paulo, São Paulo, Brazil) and were solubilized in methanol:water (8:2, *v/v*).

The analysis was performed using Shimadzu® equipment, with LC-20AT Shimadzu pumps, an automatic injector (Proeminence SIL-20AC), a C-18 column Kinetex (250 × 4.6 mm, 5 μm particle size, Phenomenex) coupled to a C-18 pre-column, a column oven (CTO-AS/VP), a UV-Vis detector (SPC-20A), and a system controller (CBM-20A). The equipment was controlled using the software LC-Solutions.

The mobile phase used was acidified water (1% acetic acid) and acetonitrile (4:6) in isocratic mode. Each run injected 15 μL of the sample in a 0.1 mL/min flow at 35 °C. A calibration curve was constructed for the quantification with all the analytical standards. All analyses were performed in quintuplicates and the analytical curves $R^2$ were > 0.99.

### 2.4. Experimental Design

As described by Faria e Souza [11], the treatments were performed orally using 0.5 mL of distilled water as a vehicle. The animals were randomly distributed into six groups (*n* = 5 per group), and dyslipidemia was induced in all of them with daily administration of saturated coconut fat (2 mL, Império dos Cocos Co., Belo Horizonte, Minas Gerais, Brazil) over 40 days. During these 40 days, the groups were treated as follows:

OSI: Treated with Ormona® SI at 200 mg/kg/day;
ORC: Treated with Ormona® RC at 200 mg/kg/day;
SG: Treated with soybean germ only at 200 mg/kg/day;
Simv.: Positive control, treated with simvastatin at 20 mg/kg/day;
Simv. + OSI: Treated with Ormona® RC (200 mg/kg/day) plus simvastatin (20 mg/kg/day);
DW: Negative control, treated only with distilled water.

#### 2.4.1. Biochemical Assay

After the 40 treatment days, the animals were kept fasted for 12 h and anaesthetized with intraperitoneal sodium thiopental at 45 mg/kg (Cristália, Co., São Paulo, São Paulo, Brazil). Blood samples were gathered (1.5 mL) from the ocular plexus, then centrifuged over 10 min at 5000 rpm. The following biochemical parameters were assessed: alanine transaminase (ALT), aspartate transaminase (AST), total cholesterol (TC) and fractions (LDL, HDL), triglycerides (TG), bilirubin, urea, and creatinine. All parameters were assessed using LabTest kits and an automatic biochemical analyzer BS 380 (Mindray Biomedical Electronics Co., Shenzhen, China).

#### 2.4.2. Aorta Scanning Electron Microscopy

On the 41st day, the animals were euthanized, and the isolation of the aorta was performed from the aortic arch to the iliac bifurcation; the thoracic region was sectioned into parts of 0.5 cm and analyzed by scanning electron microscopy (SEM-Hitachi Model-TM3030 PLUS, Tokyo, Japan) to detect atherogenesis.

#### 2.4.3. Statistical Analysis

All data are represented as a mean ± standard deviation. One-way ANOVA was used to assess statistical divergences among all groups, followed by the Tukey test, in case of statistical differences ($p < 0.05$). All analyses were performed with the software Graphpad Prism (7.0).

### 2.5. In Silico Analysis

Previously, Batista et al. [12] performed an in silico study with the compounds found in Chronic®—geranylgeraniol and tocotrienols. Considering that Ormona® also has isoflavones in its composition, an in silico analysis was conducted with the identified compounds daidzein, glycitein, and genistein.

First, a ligand structure prediction of activities and mechanisms that could be involved in the anti-dyslipidemic activity was performed using the server PASS (Prediction of Activity Spectra for Substances) [13–16].

Based on the prediction, molecular docking was performed with the target predicted to be involved in its mechanism—the human estrogen receptor (PDB ID: 1QKM). The coordinates used were 22.4, 8.04, and 113.59 (x, y, z). The docking was validated and performed using the software GOLD, as described by Matias Pereira [6].

## 3. Results

### 3.1. Chemical Analysis

The CG analysis showed that, from the annatto part of the formulations (42.6% *w/w*), 10% are total tocotrienols (γ and δ) and 28% is geranylgeraniol. From the isoflavones-rich fraction (20.8% *w/w*), the HPLC analysis showed in OSI, 1.08 % was genistein (from dry fraction weight), 0.16 % was genistin, 3.84 % was daidzein, and 3 % was glycitein. In ORC the following were identified: daidzein at 5.4%, glycitein at 2.3%, genistein at 3.6% (Figure 1). Ten percent (*w/w*) of the nutraceuticals are from a standardized extract of açaí that has been characterized before [9]; about 60% of this extract is composed of anthocyanins (mainly cyanidin 3-O-glucoside).

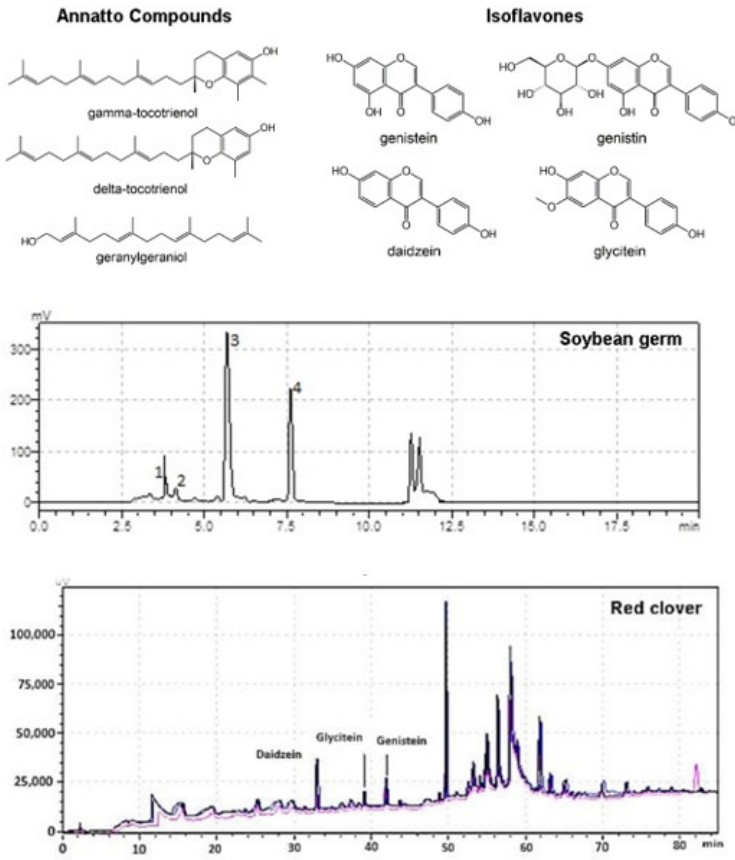

**Figure 1.** Top: compounds found in the used formulations. The first chromatogram show isoflavones identification from the soybean germ, which are the same added to Ormona® SI; the peaks represent genistein (1), genistin (2), daidzein (3), glycitein (4). The second chromatogram shows the compounds from Ormona® RC, where daidzein, glycitein, and genistin could be identified.

### 3.2. Total Bilirubin

The groups' mean values of bilirubin ± SD are shown in Table 1. There was a statistically significant difference among all groups ($p < 0.0001$; F(5,24) = 42.03). Multiple comparisons Tukey tests revealed that all groups had highly significant lower values ($p < 0.0001$) than the negative control group treated only with distilled water. Only OSI had a statistically higher bilirubin level than the positive control treated with simvastatin ($p < 0.05$). The data is summarized in Figure 2.

**Table 1.** Biochemical results (Mean ± SD).

|  | OSI | ORC | SG | Simv. | Simv. + OSI | DW |
|---|---|---|---|---|---|---|
| Total bilirubin (mg/dL) | 0.107 ± 0.005 | 0.098 ± 0.01 | 0.092 ± 0.016 | 0.084 ± 0.013 | 0.07 ± 0.007 | 0.167 ± 0.015 |
| Total cholesterol (mg/dL) | 71.75 ± 7.27 | 70.8 ± 5.76 | 70.4 ± 3.57 | 78.0 ± 16.31 | 83.4 ± 10.11 | 249.66 ± 190.89 |
| LDL (mg/dL) | 4.75 ± 1.70 | 4.20 ± 1.30 | 3.40 ± 0.54 | 5.6 ± 1.81 | 5.25 ± 2.21 | 22.33 ± 8.96 |
| HDL (mg/dL) | 47.5 ± 4.5 | 46.5 ± 5.19 | 49.00 ± 4.58 | 50.4 ± 9.15 | 53.80 ± 6.68 | 30.33 ± 2.88 |
| Triglycerides (mg/dL) | 100.25 ± 25.23 | 142.20 ± 64.80 | 134.40 ± 30.62 | 150.80 ± 56.25 | 121.20 ± 32.55 | 1028.33 ± 460.39 |
| Glucose (mg/dL) | 95.00 ± 11.04 | 102.6 ± 6.34 | 109.00 ± 5.70 | 110.4 ± 8.73 | 119.00 ± 8.68 | 391.00 ± 50.26 |
| AST (U/L) | 93.10 ± 4.49 | 87.66 ± 6.83 | 98.42 ± 16.42 | 102.96 ± 17.55 | 82.84 ± 8.99 | 129.80 ± 32.82 |
| ALT (U/L) | 42.35 ± 4.01 | 39.26 ± 6.68 | 48.23 ± 23.31 | 54.22 ± 18.82 | 39.96 ± 9.13 | 63.60 ± 43.98 |
| Urea (mg/dL) | 36.50 ± 3.69 | 37.60 ± 2.40 | 39.20 ± 4.38 | 41.40 ± 4.50 | 44.40 ± 3.64 | 80.33 ± 29.02 |
| Creatinine (mg/dL) | 0.65 ± 0.27 | 0.64 ± 0.032 | 0.642 ± 0.294 | 0.618 ± 0.022 | 0.610 ± 0.044 | 1.153 ± 0.26 |

### 3.3. Total Cholesterol

The groups' mean values of total cholesterol ± SD are shown in Table 1. There was a statistically significant difference among all groups ($p = 0.0073$; F(5,24) = 4.16). Multiple comparisons Tukey tests revealed that all groups had statistically lower TC values ($p < 0.05$) than the negative control group. There was no statistical difference between the simvastatin-treated group and the others, except for the negative control (Figure 2).

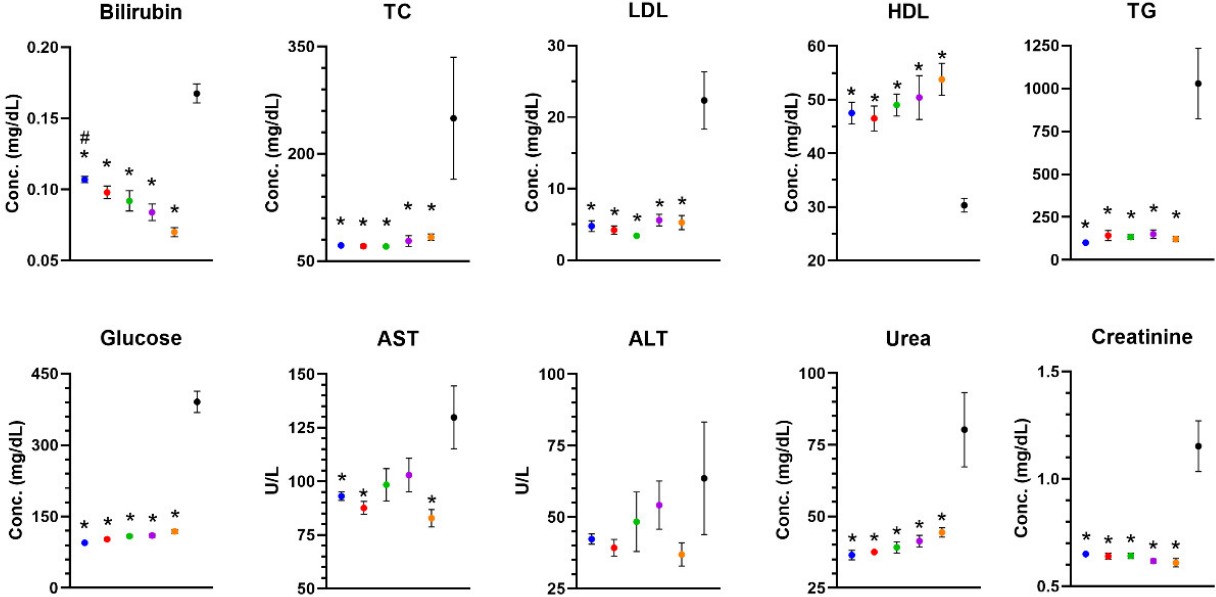

**Figure 2.** Results of the biochemical parameters (mean ± SEM). * represents statistical difference compared to the negative group treated with distilled water; # represents statistical difference compared to the positive group treated with simvastatin ($p < 0.05$; ANOVA followed by Tukey test).

### 3.4. Low-Density Lipoproteins

The groups' mean values of LDL are shown in Table 1. There was a statistically significant difference among all groups ($p < 0.0001$; F(5,24) = 16.96). Multiple comparisons Tukey tests revealed that all groups had statistically lower LDL values ($p < 0.0001$) than the negative control group. There was no statistical difference between the simvastatin-treated group and the others, except for the negative control (Figure 2).

### 3.5. High-Density Lipoproteins

The groups' mean values of HDL are shown in Table 1. There was a statistically significant difference among all groups ($p < 0.0001$; F(5,24) = 9.869). Multiple comparisons Tukey tests revealed that all groups had statistically higher HDL values than the negative control group, with $p < 0.01$ for OSI and ORC, $p < 0.001$ for SG and simv., and $p < 0.0001$ for simv. + OSI. There was no statistical difference between the simvastatin-treated group and the others, except for the negative control (Figure 2).

### 3.6. Triglycerides

The groups' mean values of TG are shown in Table 1. There was a statistically significant difference among all groups ($p < 0.0001$; F(5,24) = 18.23). Multiple comparisons Tukey tests revealed that all groups had statistically lower TG values than the negative control group, with $p < 0.0001$ for all groups. There was no statistical difference between the simvastatin-treated group and the others, except for the negative control (Figure 2).

### 3.7. Glucose

The groups' mean values of glucose are shown in Table 1. There was a statistically significant difference among all groups ($p < 0.0001$; F(5,24) = 140.9). Multiple comparisons Tukey tests revealed that all groups had statistically lower glucose levels than the negative control group, with $p < 0.0001$ for all groups. There was no statistical difference between the simvastatin-treated group and the others, except for the negative control (Figure 1).

### 3.8. Aspartate Transaminase

The groups' mean values of AST are shown in Table 1. There was a statistically significant difference among all groups ($p = 0.0043$; F(5,24) = 4.614). Multiple comparisons Tukey tests revealed that OSI's AST levels were statistically lower than the negative control group ($p < 0.05$), while in ORC and simv. + OSI the difference was more significant ($p < 0.01$). There was no statistical difference between the simvastatin-treated group and the others, except for the negative control (Figure 2).

### 3.9. Alanine Transaminase

The groups' mean values of ALT are shown in Table 1. There was not a statistically significant difference among all groups ($p = 0.4228$; F(5,24) = 1.030), as observed in Figure 2.

### 3.10. Urea

The groups' mean values of urea are shown in Table 1. There was a statistically significant difference among all groups ($p < 0.0001$; F(5,24) = 9.236). Multiple comparisons Tukey tests revealed that the urea levels from all groups were statistically lower than the negative control group ($p < 0.001$, except simv. + OSI: $p < 0.01$). There was no statistical difference between the simvastatin-treated group and the others, except for the negative control (Figure 2).

### 3.11. Creatinine

The groups' mean values of creatinine are shown in Table 1. There was a statistically significant difference among all groups ($p < 0.0001$; F(5,24) = 18.13). Multiple comparisons of Tukey tests revealed that all groups had significantly lower creatinine levels than the

negative control ($p < 0.0001$). There was no statistical difference between the simvastatin-treated group and the others, except for the negative control (Figure 2).

### 3.12. Atherogenesis Formation

The high-fat consumption could induce the formation of atherogenesis in the thoracic aorta, as observed in Figure 3. However, no atheroma plaques were observed in groups treated with Ormona® or in the positive control treated with simvastatin. The treatment with soybean germ alone failed in preventing atherogenesis.

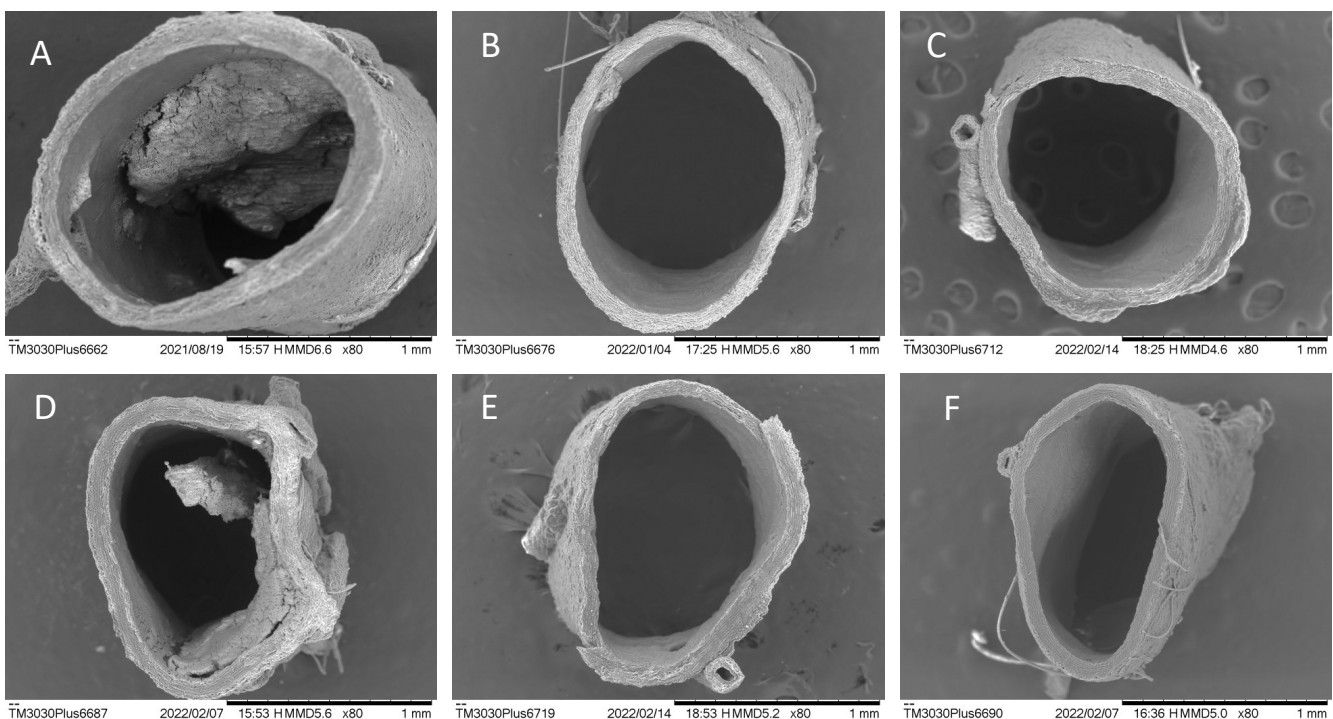

**Figure 3.** Representative SEM photomicrographs of the thoracic aorta cross-sections. (**A**) Negative control receiving only distilled water. (**B**) Treated with Ormona SI® at 200 mg/kg/day. (**C**) Treated with Ormona RC® at 200 mg/kg/day. (**D**) Treated with soybean germ only at 200 mg/kg/day. (**E**) Treated with Ormona RC® (200 mg/kg/day) plus simvastatin (20 mg/kg/day). (**F**) Positive control, treated with simvastatin at 20 mg/kg/day.

### 3.13. In Silico Results

The outputs given by PASS are shown in Table 2, where Pa means the probability of activity and Pi means the probability of inactivity. All three molecules had a similar set of activities predicted and similar Pa values, which is expected due to their structural similarity.

PASS indicated estrogen receptor agonism for all of the molecules; hence, we performed molecular docking with this target. The RMSD value of validation was 0.391 Å, which is within the accepted limit of 2 Å [17]. The structure ligand was validated with a score of 78.31. Genistein's docking score was 79.41 and performed 9 interactions with the target's active site; daidzein's score was 79.11 and performed 13 interactions; glycitein's score was 69.94 with 16 interactions; formononetin scored 70.96 with 11 interactions and biochanin A scored 72.90 with 12 interactions. The best docking poses with these interactions and the amino acid residues involved are depicted in Figure 4.

**Table 2.** Prediction of Activity Spectra for Substances (PASS), where Pa means the probability of activity and Pi means the probability of inactivity.

| Molecule | Pa | Pi | Activity predicted |
|---|---|---|---|
| **Daidzein** | 0.595 | 0.006 | Menopausal disorders treatment |
| | 0.562 | 0.004 | Estrogen receptor agonist |
| | 0.582 | 0.009 | Bone disorders treatment |
| **Genistein** | 0.560 | 0.009 | Menopausal disorders treatment |
| | 0.583 | 0.004 | Estrogen receptor agonist |
| | 0.584 | 0.009 | Bone disorders treatment |
| **Glycitein** | 0.592 | 0.007 | Menopausal disorders treatment |
| | 0.469 | 0.005 | Estrogen receptor agonist |
| | 0.474 | 0.017 | Bone disorders treatment |
| **Formononetin** | 0.584 | 0.007 | Menopausal disorders treatment |
| | 0.482 | 0.005 | Estrogen receptor agonist |
| **Biochanin A** | 0.549 | 0.009 | Menopausal disorders treatment |
| | 0.522 | 0.005 | Estrogen receptor agonist |
| | 0.519 | 0.013 | Bone disorders treatment |

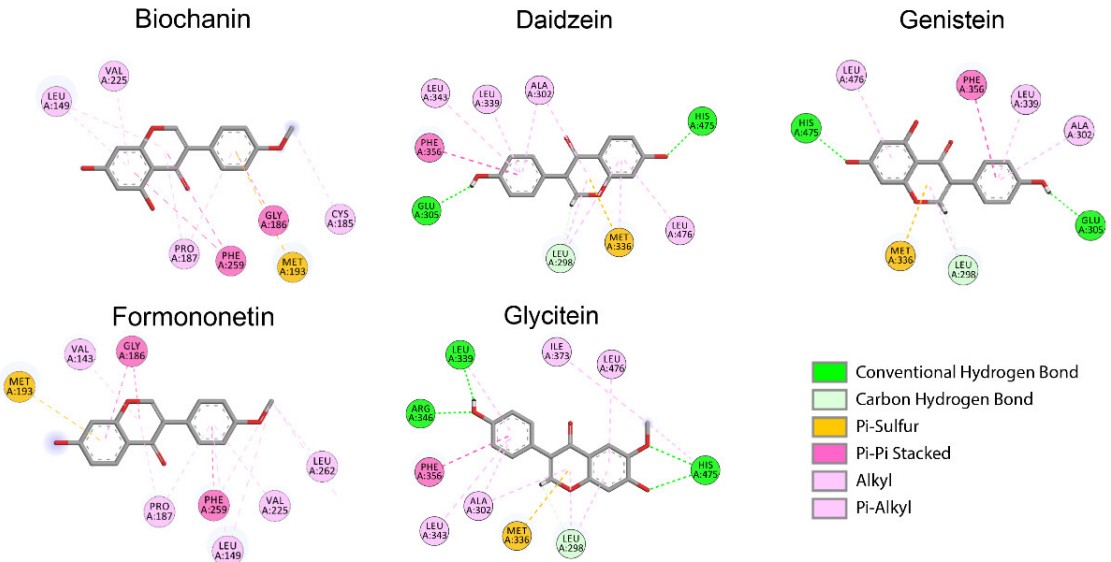

**Figure 4.** *Docking* poses predicted with estrogen receptor (PDB ID 1QKM).

## 4. Discussion

Around 40% of OSI and ORC are annatto (*Bixa orellana*) compounds, with the same composition as Chronic® [6]. Of this 40%, ~10% are tocotrienols (mainly δ) and ~28% are geranylgeraniol. In OSI, 20.8% are composed of soybean germ, while in ORC they are composed of red clover extract (*Trifolium pratense*). Finally, 10% of the formulations are of an anthocyanin-rich (mainly cyanidin 3-O-glucoside) standardized extract of açaí (*Euterpe oleracea*).

The isoflavones content of the soybean germ and OSI was composed mainly of daidzein and glycitein; genistein and genistin were identified in a lower quantity (Figure 1). These compounds are in accordance with the literature on soybean germ composition [18]. As for ORC isoflavones content, daidzein, glycitein, and genistein could be identified, but several other compounds appeared in a high quantity (Figure 1).

According to the literature, some of these unidentified peaks could be other isoflavones characteristic of *T. pratense*, mainly formononetin and biochanin A [19,20]. These compounds are structurally similar to 17β- estradiol and, after demethylation by cytochromes P450 in the intestine and liver, will be converted into daidzein and genistein, respectively [21].

After assessing their composition, we tested these formulations against high-fat-induced dyslipidemia. Abnormally increased blood lipid levels play a pivotal role in the development of cardiovascular and metabolic diseases, such as diabetes and atherosclerosis [22–26], and hence, it represents a public health burden.

A popular model to induce dyslipidemia is through the daily administration of coconut oil (*Cocus nucifera*). Coconut oil consumption can increase cholesterol and triglycerides synthesis, modeling an abnormal metabolic condition [27]. Our study used the same oil as that from [28], which revealed through GC-MS that it was composed of lauric acid (57.8%), myristic acid (17.3%), and palmitic acid (9.4%), among others, totaling 92.64% of saturated fat.

The results show a massive increase in TC, LDL, and TG levels and decreased HDL levels in untreated animals (Figure 2). The treatment with OSI, ORC, and SG had very similar results in this aspect; all of them prevented the increase of TC, LDL, and TG levels and the de-crease of HDL, similarly to simvastatin.

Both tocotrienols and geranylgeraniol are known inhibitors of HMG-Coa Reductase-a major enzyme involved in cholesterol synthesis [29,30], which could in part explain the lipid-lowering effect of OSI and ORC. In fact, the original Chronic®, in which there is no addition of isoflavones, significantly improved the blood lipid profile in testosterone deficiency-induced dyslipidemia [6]. Recently, Batista et al. [12], based on structure similarity and docking studies, proposed that other lipid-lowering mechanisms could be involved in annatto compounds, such as squalene monooxygenase inhibition and lanosterol synthase inhibition.

Considering that the animals treated only with soybean germ also had significantly different results than the untreated group, their role in Ormona® SI and RC cannot be ruled out. In fact, isoflavones also are known to decrease lipid levels and ameliorate menopausal symptoms [8,22]. The mechanism whereby these compounds act is different from geranylgeraniol and tocotrienols. While the latter act mainly through the decreased activity of HMG-CoA Reductase, isoflavones are believed to inhibit $7\alpha$-hydroxylase [8], but the activation of estrogen receptors (ER) could also have a role [19].

We looked for mechanisms that could explain the blood lipid improvement using in silico methods. Based on the structures, PASS predicted that all of the isoflavones could act as agonists of the estrogen receptor. Hence, we performed molecular docking, which corroborated this hypothesis with docking scores similar to the crystallized molecule in the protein structure used. Formononetin and biochanin A had considerably lower scores compared to the other compound, but, as mentioned, these compounds are metabolized to daidzein and genistein.

Agonism of estrogen receptors will cause its dimerization and further inhibition of sterol regulatory element-binding protein 1 (SREBP1), inhibition of peroxisome proliferator-activated receptor $\gamma$ (PPAR$\gamma$), and inhibition of CCAAT/enhancer-binding protein (C/EBP). Inhibition of these pathways will decrease lipid accumulation [19]. Moreover, the activation of the estrogen receptor regulates LDL receptors and, consequently, LDL metabolism. This could explain why low estrogen levels are linked to increased LDL cholesterol [31].

Besides, by drastically decreasing the abnormal lipid levels, the treatments prevented some of its complications. It is known that the specific organ for fat storage is the adipose tissue; however, in obese states, there is lipids accumulation in ectopic sites, including the liver, kidney, skeletal muscle, and possibly β-pancreatic cells [32]. The increased concentration of fatty acids in the muscles and liver is directly linked to insulin resistance [32,33]. In accordance, our results showed that the untreated animal had abnormally high glucose levels, which were prevented by the treatments as well (Figure 2).

Besides insulin resistance, fat deposition in the liver can cause increased inflammatory states, decreased autophagy, endoplasmic reticulum stress, mitochondrial dysfunction, and death of hepatocytes. Here, this hepatotoxicity was observed, evidenced by in-creased bilirubin levels and AST. ALT did not increase statistically significantly, but a clear trend can be observed (Figure 2). Curiously, only the treatments with Ormona® de-creased

AST levels in a statistically significant manner, with better results than soybean germ or simvastatin alone.

Ectopic lipid accumulation in the kidney can induce dysfunction and increase inflammation, hampering glomerular filtration [34]. This kidney dysfunction was observed here, evidenced by abnormal urea and creatinine concentrations, which could be prevented by the treatments (Figure 2).

The excessive increase of LDL can cause its deposition in the arteries' inner layer, the tunica intima. This event will trigger a complex cascade that will cause inflammation, ROS formation, macrophage infiltration, and, consequently, the formation of foam cells, leading to atherogenesis and, eventually, atherosclerosis—which is linked to increased morbidity and mortality [28]. The SEM analysis showed atheroma plaques, mainly in the negative control, as shown in Figure 3. It is curious that despite presenting a lipid-lowering capacity, the group treated with only soybean germ had atheroma formation; this indicates that the anti-atherogenesis activity of Ormona® may be at least in part independent of its hypocholesterolemic effect.

Possibly, the anthocyanins from the Ormona® composition can explain their superior effect compared to the isoflavones alone in preventing atherogenesis. This class of compounds will act in several ways to hamper this process. According to the literature, anthocyanins can prevent the activation and recruitment of neutrophils and monocytes, decrease the expression of adhesion molecules necessary for the infiltration of inflammatory cells in the endothelium (e.g., ICAM-1, VCAM-1), and decrease the release of other pro-inflammatory mediators [27].

This study tested the effect of two new nutraceuticals with annatto, açaí compounds, and isoflavones from two sources. Overall, these two formulations had similar results for their efficacy. Although the isoflavones alone could similarly improve the lipid profile, OSI and ORC had better protection from lipid-induced hepatoxicity, as evidenced by the AST and ALT results, and prevented atherogenesis. Some limitations of the study are the lack of more mechanism-targeted assays to corroborate the putative mechanisms of action proposed. Additionally, although we presume that the unidentified peaks in ORC and OSI are formononetin and biochanin A, based on the literature for red clover, this cannot be 100% affirmed due to the lack of certified standards for all of the molecules. It is important to also keep in mind that the two formulations are mixtures of compounds, hence it is difficult to precisely know which molecules are responsible for each activity, and a possible synergism among them cannot be ruled out.

## 5. Conclusions

This study tested the effect of two new nutraceuticals with annatto compounds (geranylgeraniol and tocotrienols), açaí compounds (cyanidin-3-O-glucoside), and isoflavones from two sources (soybean germ in OSI and red clover in ORC). Both nutraceuticals improved the high-fat-induced abnormal lipid profile significantly, with values close to the control compound, simvastatin. These products could also prevent dyslipidemia complications, such as abnormal glucose levels, hepatotoxicity, nephrotoxicity, and atherogenesis. Moreover, co-treatment with the nutraceuticals and simvastatin could protect against liver lipotoxicity with better results than with simvastatin alone.

## 6. Patents

A patent deposit resulting from this work was requested at the National Patent and Innovation Institute of Brazil (INPI) with the number BR-10 2022 008408-4.

**Author Contributions:** Conceptualization, J.C.T.C.; methodology, A.P.S.R., R.d.S.B., N.N.D.C., M.A.B., H.R.d.S. and L.I.d.S.H.-M.; validation, M.O.d.S.S. and P.F.S.; formal analysis, A.M.F.; writing—original draft preparation and results discussion, A.C.M.P.; writing—review and editing, data analysis, J.C.T.C.; supervision, J.C.T.C.; project administration, J.C.T.C.; funding acquisition, J.C.T.C. All authors have read and agreed to the published version of the manuscript.

**Funding:** This research was funded by Coordenação de Aperfeiçoamento de Pessoal de Nível Superior, grant number 1723/2018-00 PROCAD-AMAZONIA, and Conselho Nacional de Desenvolvimento Científico e Tecnológico, grant Proc. 403587/2020-4.

**Institutional Review Board Statement:** The authors followed the guidelines for animal experimentation established by the National Council for Animal Experimentation of Brazil (CONCEA). The Ethics Committee approved the project on the Use of Animals (CEUA) of the Federal University of Amapá-UNIFAP under protocol number 003/2021.

**Data Availability Statement:** Data is contained within the article.

**Acknowledgments:** The first author was awarded a student scholarship from Capes. Funding was provided by CAPES [1723/2018-00 PROCAD-AMAZONIA], CNPq [Proc.403587/2020-4], and INCT North-Northeast Network of Phytoproducts—CNPq.

**Conflicts of Interest:** The authors declare that they have no known competing financial interests or personal relationships that could have appeared to influence the work reported in this paper.

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
