# Peer review of "Ormona® SI and Ormona® RC—New Nutraceuticals with Geranylgeraniol, Tocotrienols, Anthocyanins, and Isoflavones—Decrease High-Fat Diet-Induced Dyslipidemia in Wistar Rats"

_nutraceuticals, doi:10.3390/nutraceuticals2040024_

Round 1

Reviewer 1 Report

Comments and Suggestions for Authors

Reviewing comments

In this paper, authors have planned to evaluate the efficacy of Ormona® SI (OSI) and Ormona® 55 RC (ORC), 2 nutraceutical components extracted from Bixa orellana, against dyslipidemia largely reported to be linked to a higher risk of cardiovascular diseases.

First of all, authors have characterised the extracts to show the overall composition of the both OSI and ORC, evaluated the efficacy of these two extracts on various lipid biomarkers of the female Wistar rats, and finally the used bioinformatics tools to study the molecular docking to highlight potential targets to be involved in their mechanisms.

The paper is well constructed, well written and highly illustrated, and results are very interesting, and can be potentially accepted. However, there are some points that should be addressed.

1. In the Introduction part (line 37) authors have to specify for each marker low/high level associated with dyslipedmia.

2. The main objective of the present study is to evaluate the efficacy of these nutraceuticals on dyslipidemia. Thus, it will be interesting to add another paragraph to mention how these abnormalities are currently managed.

3. In Material and Methods line 106, is it 250°C or 290°C?

4. In the in vivo experimental design, it was very confusing. Authors said that dyslipidemia was induced in all groups by administering 2 mL of saturated coconut fat over 40 days (lines 133-134). Rats were then divided to 6 groups and were subjects to different treatments for 40 days. Did both the dyslipidemia induction and the intervention needed 40 days each?

5. The first paragraph of Results section showed that OSI and ORC extracts have the same composition in terms of oil compounds, total tocotrienols (γ and δ tocotrienols), geranylgeraniol and isoflavones-rich fractions. It’s surprising!

6. Lines 186-188, authors showed that there’s a significant difference between ORC (0.098) and SG (0.092) but not between ORC (0.098) and “Simv. + OSI” (0.07), please verify!

7. Many sentences in the Results part are more likely to be moved to the Discussion part, e.g. lines 219-221: According to the literature… (n=10)[16]. Idem for lines 203-232, 241-243, 252-253 and 262-263.

8. Overall, most results regarding blood analyses are given in the text and illustrated in the Figure 2. I think that giving these results in a Table will be more visible and easy to fellow.

9. The tittle of Figure 1 must be re-edited giving a tittle an adequate tittle and specify A for chemical structures of compounds and B for obtained chromatograms.

10. The Discussion part have to give the limitations of the study, especially that the obtained activities are most likely attributed to the mixture and it’s widely reported that there’s many synergic activities between the compounds that are responsible of the obtained results.

Minor points.

1. Lines 57 – 79 are given in duplicate and must be deleted.

2. Lines 115-116, the sentence must be rephrased as: 100 mg of the product was mixed with 40 mL of a solution of bi-distilled water

3. There are some typological errors such:

- We don’t use coma before and

- Line 153, 41st, please make “st” in superscript       

- We never start a sentence with a number

- Latin words, like in silico and in vivo, must be in italic

Please check the paper and make the requested corrections

Author Response

Comments on corrections are in the attached document.

Reviewer 2 Report

Comments and Suggestions for Authors

After carefully reading, the present manuscript, the following are my remarks. In general current work provide useful data on the topic the authors dealt which is mainly focused on the biological part. Nevertheless, the authors have connected their research conclusions to analytical experimentation concerning the tocopherols content as well as isoflavones quantification of the tested material. Unfortunately, this part has not been given clearly enough while some information should be given at least as supporting information. In more detail:

What was the sample/es that the authors used for the GC-MS analysis? Did they inject directly portions from the neutraceuticals used for their bioassays? If this is the case, then since these are not volatiles entirely on what basis authors expressed the percentages mentioned in Lines  176-178?

Concerning the isoflavones quantification did the procedure take place as a triplicate, as it should, according to analytical protocols?

Lines 128-129. Did the authors construct one calibration curve for each authentic compound separately or a universal one? Please provide R2 values for the curves mentioned.

Lines 178-184. What do percentages, given for the detected isoflavones, express?

Finally, Lines 58-79 should be deleted. 

Author Response

(The authors gave the same response as above.)

Reviewer 3 Report

Comments and Suggestions for Authors

This work evaluated the effect of two new nutraceuticals with annatto compounds (geranylgeraniol and tocotrienols), açaí compounds (cyanidin-3-O-glucoside), and isoflavones from two sources (soybean germ in OSI and red clover in ORC) against high-fat-induced dyslipidemia in female Wistar rats. This study brings novelty to the field and are enriched in experiments resulting in fruitful findings. However, here are some remarks before recommendations for publication:

-          In the text, the name of the plants should be written in italics;

-          At the end of the Introduction, I suggest adding a paragraph explaining clearly the objective and the aim of the work;

-          The introduction is repeated twice in material and methods section;

-          In line 92, please correct it is kindly provided instead of “Kindle provided”;

-          In lines 115/146: please reword the sentence and avoid starting it with a number;

-          In line 147: There is a punctuation mistake;

-          In materiel and methods and discussion sections, please check your reference style in the text, sometimes you use numbers along with citations and sometimes you only use citations alone;

-          In figure 1 (TOP), try to make the compounds names clear with better quality, it is blurry;

-          In references section, the numbers of references are repeated twice.

Author Response

(The authors gave the same response as above.)

Round 2

Reviewer 2 Report

Comments and Suggestions for Authors

In the present revised form, the manuscript could proceed for publication.